# Study of the Antioxidant and Antibacterial Effects of *Genipa americana* L. Against Food Pathogens

**DOI:** 10.3390/foods14060944

**Published:** 2025-03-10

**Authors:** Lariza Leisla Leandro Nascimento, Geane Gabriele de Oliveira Souza, Ana Cecília Calixto Donelardy, Maria Inácio da Silva, Cícera Alane Coelho Gonçalves, Ana Maria Duarte Fernandes, José Walber Gonçalves Castro, Débora Odília Duarte Leite, Irwin Rose Alencar de Menezes, José Galberto Martins da Costa, Fabíola Fernandes Galvão Rodrigues

**Affiliations:** 1Postgraduate Program in Chemical Biology, Departament of Chemical Biology, Universidade Regional do Cariri, Coronel Antônio Luíz Street, 1161–Pimenta, Crato 63105-010, Ceará, Brazil; lariza.leandro@urca.br (L.L.L.N.); geane.souza@urca.br (G.G.d.O.S.); cecilia.donelardy@urca.br (A.C.C.D.); maria.i.silva@urca.br (M.I.d.S.); alane.goncalves@urca.br (C.A.C.G.); ana.fernandes@urca.br (A.M.D.F.); josewalber.castro@urca.br (J.W.G.C.); biodeboraleite@yahoo.com.br (D.O.D.L.); irwin.alencar@urca.br (I.R.A.d.M.); fabiolafer@gmail.com (F.F.G.R.); 2Natural Products Research Laboratory, Universidade Regional do Cariri, Coronel Antônio Luíz Street, 1161–Pimenta, Crato 63105-010, Ceará, Brazil; 3Northeast Biotechnology Network–RENORBIO, UniversidadeEstadual do Ceará, Av. Dr. Silas Munguba Ave., 1700–Campus do Itaperi, Fortaleza 60714-903, Ceará, Brazil

**Keywords:** *G. americana*, chemical profile, enterobacteria, antioxidants, foodborne pathogens

## Abstract

This study presents the chemical profile of the ethanolic extract of *Genipa americana* L. stem bark and the evaluation of its antibacterial and antioxidant activities. The chemical prospecting consisted of a qualitative analysis and quantification by HPLC-DAD. An antibacterial evaluation was performed using broth microdilution to determine the MIC, while gentamicin and amikacin were used to modify the antimicrobials. The antioxidant tests included the DPPH^•^ method, ABTS^•+^ radical cation capture, Fe^2+^ chelation, Fe^3+^ reduction, and oxidative degradation of deoxyribose. Phytochemical tests identified its flavonoid and alkaloid classes, and an HPLC analysis allowed for caffeic acid quantification in the extract. The results of this study showed satisfactory MICs for *E. coli* and *K. pneumoniae*, 256 µg/mL; *S. flexneri* and *P. vulgaris*, 512 µg/mL; and *S. typhimurium*, ≥ 1024 µg/mL. Furthermore, there was a modifying effect on the bacterial strains, except for *S. enterica*. The antioxidant tests using the DPPH^•^ method showed an IC_50_ of 298.1 µg.mL^−1^, with the highest percentage of ABTS^•+^ radical cation capture occurring at a concentration of 500 µg/mL; regarding Fe^2+^, chelating activity was not present, and for Fe^3+^ reduction, the best concentrations were 10 µg/mL and 25 µg/mL. The data obtained can be used to turn *G. americana* into a viable species as an agent for antibacterial and antioxidant functionalities in foods.

## 1. Introduction

Species from the Rubiaceae family have become increasingly important in pharmacological and biological studies due to the various classes of secondary metabolites, such as alkaloids, tannins, saponins, terpenes, and flavonoids, which are associated with their biological activities [1].

*Genipa americana* L., popularly known as “jenipapo”, belongs to the Rubiaceae family and has a vast chemical composition, among which iridoids are the most important metabolites described [2,3]. Popular knowledge and ethnopharmacological studies on the species suggest significant antimicrobial activities [4], and pharmacological studies conducted on the leaves of *G. americana* show anti-inflammatory, antiangiogenic, antidiarrheal, antisyphilitic, and antidiabetic properties and anticonvulsant effects [2]. Research into the chemical composition and/or biological activity of fruits of the species has resulted in the isolation and characterization of secondary metabolites, monoterpenes, and iridoids [5].

Literature reports indicate that “jenipapo” has phenolic compounds in its chemical composition, which are related to potential antioxidant activities, as well as substances which protect the human body and act against oxidative stress, preventing the development of chronic degenerative diseases. Natural antioxidants are an alternative that is attracting interest due to their safe use and therapeutic and nutritional effects in food [6,7]. According to ethnobotanical surveys, boiled “jenipapo” leaves are used in cases of intestinal decompensation, proving to have antidiarrheal activity [2].

Due to the presence of metabolites capable of inhibiting bacterial growth, investigations are being conducted into the Enterobacteriaceae, a family of Gram-negative bacteria involved in a high number of hospital infections and health care needs, and which have a long life cycle and develop antibiotic resistance. Species from this family are involved in food poisoning and diarrheal episodes [8,9].

The compounds present in these species and popular reports show their potential to reduce the risk of food contamination by pathogens and to reduce food oxidation by avoiding the action of free radicals, guaranteeing consumer safety [7,10]; as a result, this study describes the chemical profile, and antioxidant and antibacterial potential against enterobacteria of a “jenipapo” stem bark extract. This study aims to identify the chemical composition and determine the antibacterial activity against enterobacteria and the antioxidant profile of the ethanolic extract of genipap stem bark.

## 2. Materials and Methods

### 2.1. Plant Tissues and Extract Production

*Genipa americana* L. stem barks were collected in Serra Olho D’água, municipality of Jardim, state of Ceará, Brazil, during the month of June 2023. The plant tissue was identified under number 16,048 by the Dárdano de Andrade Lima Herbarium, which belongs to Regional University of Cariri. The fresh bark (950 g) was macerated in ethanol for 72 h. Subsequently, the solution obtained was filtered and distilled in a rotary evaporator under reduced pressure and controlled temperature between 40 and 50 °C, with a yield of 2.80% (*m*/*m*).

#### 2.1.1. Chemical Identification

Chemical prospecting was performed following the methodology of Matos [11] and Simões et al. [12] to identify the classes of secondary metabolites present in the ethanolic extract of *G. americana* bark. The result of the test is qualitative and consists of visual observation through color change or precipitate formation after the addition of specific reagents to the sample solutions.

#### 2.1.2. HPLC/DAD

An HPLC analysis of phenolic acids and flavonoids was performed using an Agilent 1260 HPLC system (Agilent Tech., Waldbronn, Germany) equipped with a UV-Vis DAD. The separation was carried out following the gradient method using a C18 chromatographic column (250 mm × 4.0 mm × 5 µm, Macherey-Nagel, Düren, Germany). Phenolic standard solutions and mixtures were injected into the system using an autoinjector. The mobile phase consisted of a solvent mixture of A (ultrapure water) and B (methanol:acetonitrile, 60:40, HPLC grade, Agilent Tech., Waldbronn, Germany), both acidified with 0.1% formic acid (Sigma Aldrich, St. Louis, MI, USA) using the following elution gradient: 0–15 min: 15% B in A; 17 min: 40% B in A; 30 min: 30% B in A; 38 min: 15% B in A; and maintaining this composition for 45 min. The wavelengths used were 310 nm for caffeic acid, p-coumaric acid, and ferulic acid; 290 nm for cinnamic acid, naringenin, and pinocembrin; and 340 nm for apigenin. The mobile phase flow rate used was 0.5 mL/min, and the injection volume was 20 µL. The mobile phases and all solutions and samples were filtered through a Millipore membrane filter with a filter diameter of 13 mm and a pore diameter of 0.22 µm (milllipore). The samples were dissolved in HPLC grade methanol (30 mg/mL). Quantification was performed by integrating peaks using the external standard method. The analyses were carried out at room temperature and in triplicate, and the peaks were confirmed by comparing their retention time with those of the reference standards and by DAD spectra (190 to 400 nm). The quantifications of the compounds were based on analytical curves of the reference standards. The limit of detection (LOD) and limit of quantification (LOQ) were calculated based on the standard deviation of the responses and the slope using three independent analytical curves. LOD and LOQ were calculated as 3.3 and 10 σ/S, respectively, where σ is the standard deviation of the response and S is the slope of the calibration curve.

### 2.2. Antibacterial Activity

#### 2.2.1. Antibacterial Evaluation and Minimum Inhibitory Concentration (MIC)

The method used to evaluate antibacterial activity was microdilution based on CLSI [13]. Standard enterobacteria bacteria were used, consisting of Gram (-) bacilli: *Escherichia coli* ATCC 25226, *Shigella flexneri* INCQS 00152, *Proteus vulgaris* ATCC 13315, *Klebsiella pneumoniae* ATCC 4352, and *Salmonella typhimurium* NCTC 12023. Serial microdilution of EEFGA was conducted, and the concentrations of the samples ranged from 1024 to 16 μg/mL. The plates were incubated for 24 h at 37 °C [13]. Reading was conducted using sodium resazurin, which indicates microbial growth [14].

#### 2.2.2. Evaluation of Modifying Activity

The test for modulating activity was conducted in the presence and absence of the natural compound, employing microdilution in triplicate and using the MIC of the ethanolic extract against antibiotics of the aminoglycoside class (gentamicin and amikacin). The amount of extract used was calculated based on the sub-inhibitory concentration (MIC/8), and the concentrations of the samples ranged from 1.024 to 1 μg/mL. The plates were incubated for 24 h at 37 °C and then read using resazurin [14].

### 2.3. Evaluating Antioxidant Activity

#### 2.3.1. DPPH^●^ Free Radical Inhibition

The free radical scavenging activity was determined using the DPPH^●^ photocolorimetric method, according to Rufino et al. [15], with modifications. Different concentrations (5, 10, 25, 35, 75, and 100 µg/mL) of the extract diluted in ethanol were prepared. In a 96-well ELISA microplate, 150 µL of each concentration was added in duplicate, along with 150 µL of the ethanolic solution of the free radical DPPH^●^ at 0.3 mM. The positive control (ascorbic acid) was made using the same procedures employed to prepare the different concentrations of the extracts. After 30 min and incubation at room temperature protected from light, absorbance measurements were taken at 518 nm in a UV-visible spectrophotometer. Antioxidant activity inhibition values were expressed in percentages (%) and calculated using Equation (1), followed by a graphical representation:(1)$$AA\%=100-\frac{\left[\left({Abs.}_{Control\;}-{(Abs.}_{Extract}-{Abs.}_{white\;}\right)\right]\times 100\;}{{Abs.}_{Control}}$$
where *Abs._Control_* is the absorbance of the DPPH^●^ solution, excluding the extract; *Abs._Extract_* is the absorbance of the extract with DPPH^●^; and *Abs._White_* is the absorbance of the extract without DPPH^●^.

#### 2.3.2. Reduction Power of Fe^3+^ Ion and Chelation of Fe^+2^

In accordance with the methodology described by Salazar et al. [16], the extraction capacity to chelate Fe^2+^ or reduce Fe^3+^ was determined through the ortho-phenanthroline assay. The assay was based on evidence of free Fe^2+^ with o-phenanthroline forming an orange-red complex called ferroin (ph) 3Fe^2+^, which was measured with an absorbance reading at 510 nm. The extracts were prepared at final concentrations that ranged from 10 to 1000 μg/mL in the well. For the test, volumes of 500 μL of the extract were mixed with 500 μL of the Fe^2+^ ion in vitro using a 1000 μM FeSO_4_ solution and with 500 μL of the Fe^3+^ ion in vitro using a 1000 μM FeCl_3_ solution, under light and refrigerated conditions. After this, 50 μL was suspended in 96-well plates containing 250 μL of the tris-HCl mix with phenanthroline pH 7.4 ± 0.1. The blank was made by replacing the tris-HCl mixture with phenanthroline with milliq water. The time taken was 2.5 min, and the reading was carried out on an Elisa spectrophotometer at 510 nm. The calculation for percentage reduction was performed according to Equation (2) and, for chelation, with Equation (3).(2)$$\mathrm{Reduc}.\;\mathrm{Power}\;{\mathrm{Fe}}^{3+}(\%)=\frac{[({Abs.}_{Extract\;{Fe}^{2+}}-{Abs.}_{Ext.\;White\;{Fe}^{3+})-{Abs.}_{Control.}{Fe}^{3+}]\;}\times 100}{{Abs.}_{Control{Fe}^{2+}}}$$(3)$$\mathrm{Chelat}.\;\mathrm{Power}\;{\mathrm{Fe}}^{2+}(\%)=\frac{[({Abs.}_{Extract\;{Fe}^{2+}}-{Abs.}_{Ext.\;White\;{Fe}^{2+})-{Abs.}_{Control.}{Fe}^{2+}]\;}\times 100}{{Abs.}_{Control\;{Fe}^{2+}}}$$
where *Abs._Control_* Fe^2+^ is the absorbance of *o*-phe with Fe^2+^ without the extract; *Abs._Control_* Fe^3+^ is the absorbance of *o*-phe with Fe^3+^ without the extract; *Abs._Extract_* Fe^3+^ is the absorbance of the extract with o-phe and Fe^3+^; and *Abs._Ext.White_* Fe^3+^ is the absorbance of the extract without *o*-phe with Fe^3+^.

#### 2.3.3. ABTS^●+^ Radical Cation Capture

The ABTS^●+^ solution has a strong blue-green color and was obtained by oxidizing the diammonium salt of ABTS (3-ethylbenzothiazoline-6-sulfonic acid) with potassium persulfate (K_2_S_2_O_8_), according to the methodology of Salazar et al. [16] with adaptations. To produce the ABTS^●+^ free radical, the ABTS solution (7 mM) was combined with potassium persulphate (140 mM) and conditioned in the dark for 16 h at room temperature. Subsequently, the ABTS^●+^ solution (7 mM) was diluted in ethanol (95%) to adjust the absorbance between 0.700 ± 0.020 and 0.800 ± 0.020; for the assay, 30 µL of EECGA (between 1000 and 10 µL) was placed into the ABTS^●+^ solution. After 6 min of reaction with the solution, readings were taken on a UV-vis spectrophotometer at a wavelength of 734 nm. For a positive control, ascorbic acid was used, and for the white absorbance, methanol was used. The test was conducted in triplicate with antioxidant activity inhibition values expressed as percentages (%) and calculated using Equation (1).

#### 2.3.4. Deoxyribose Oxidative Degradation

The method used was adapted from Puntel et al. [17]. The solution of the extracts was prepared at concentrations from 10 to 1000 µg/mL. For the test, 450 µL of potassium phosphate buffer (7.5 mM), 150 µL of deoxyribose (1.5 mM), 240 L of H_2_0_2_ (0.8 mM), 240 µL of FeSO_4_ (80 µM), and 320 µL of distilled water were added to 100 µL of the crude seed or pulp extracts diluted at different concentrations. For the white absorbance, the same mixture was prepared in the absence of deoxyribose. All samples were incubated for 60 min at 37 °C. Moreover, they received 750 µL of trichloroacetic acid (2.8%) and 750 µL of thiobarbituric acid (0.8%) and were re-incubated for 20 min in a heated bath at 100 °C. The reading was performed in a spectrophotometer at 532 nm. The results were calculated according to Equation (3). The analysis was performed in triplicate.(4)$$\frac{[1-{(Abs.}_{Control-}{Abs.}_{white})]\times 100\;}{{Abs.}_{Control}}$$

### 2.4. Statistical Analysis

Microbiological tests were analyzed by two-way ANOVA and the Bonferroni test using GranphPad Prism 6.0 software, considering results *p* < 0.05 to be statistically significant. For the toxicological activity, the results were tabulated using the linear regression model and Tukey’s test for multiple comparison.

## 3. Results

### 3.1. Chemical Profile

The chemical prospecting of an ethanolic extract from the bark of *G. americana* (EECGA) identified the following secondary metabolite classes: flavonoids (flavones, xanthones, flavonols, chalcones, aurones, catechins, and flavonones) and alkaloids, as observed in Table 1.

An analysis by HPLC-DAD for the quantification of phenolic acids and flavonoids (caffeic acid, p-coumaric acid, ferulic acid, cinnamic acid, naringenin, pinocembrin, and apigenin) demonstrated the presence of caffeic acid in the amount of 0.11446 ± 0.003873 mg/g, with the results of the values expressed as mean (mg/g of sample) ± SD (n = 3), consisting of 0.011%, demonstrated in Figure 1. This result was expected due to the preliminary quantification of phenols and flavonoids.

### 3.2. Antibacterial Activity

The results against the bacteria tested showed an inhibitory capacity for the extract with varying MICs. *E. coli* and *K. pneumoniae* showed MICs of 256 µg/mL; *S. flexneri* and *P. vulgaris* showed 512 µg/mL; and for *S. enterica*, the result was ≥1024 µg/mL. Caffeic acid presented an MIC of 128 µg/mL for *K. pneumoniae* and, for the other bacteria, ≥1024 µg/mL. For the potential modifier of the antibiotic effect by EECGA on the activity of aminoglycosides (amikacin and gentamicin), the results are shown in Figure 2.

### 3.3. Antioxidant Activity

This study evaluated the antioxidant capacity of the ethanolic extract of the stem bark of *G. americana* using the DPPH^●^ method, revealing an IC_50_ of 298.1 µg.mL^−1^ (Figure 3a). The results for the FRAP method are shown in Figure 3b,c. Figure 3d shows the results of EECGA in the capture of the ABTS^•+^ radical cation, comprising an IC_50_ of 77.25 µg.mL^−1^. According to the test for the oxidative degradation of deoxyribose, the results of this study with EECGA showed an IC_50_ of 261.9 µg/mL, the amount required to eliminate 50% of the hydroxyl radical. For the same tests, caffeic acid presented an IC_50_ of 1.46 µg/mL^−1^ for the DPPH^●^ test, an IC_50_ of 0.078 µg/mL^−1^ for the ABTS^●+^ test, an IC_50_ of 8.01 µg/mL^−1^ in the FRAP method for the chelation of Fe^2+^ ions, no 50% inhibition in the reduction of the Fe^3+^ ion, and an IC_50_ of 12.02 µg/mL^−1^ for deoxyribose.

### 3.4. Multivariate Statistical Analysis

The PCA biplot (PC1 and PC2) provides a comprehensive view of the scores and loadings, effectively illustrating how the compounds are separated along the first two principal components (Figure 4a). PC1, which explains nearly 100% of the variance, reflects significant differences in some variables, as confirmed by the dendrogram (Figure 4b). The distinct clusters formed by EECCGAL and caffeic acid further validate the thoroughness of the analysis. The loading plot, with its arrows indicating the strong positive association of PC1 with antioxidant activity (DPPH, ABTS, and deoxyribose), and the negative contribution of the iron chelating variable, adds another layer of understanding. Similarly, PC2 shows positive values for antioxidant activity (ABTS and iron chelating) and negative contributions for DPPH and desoxyribose.

A correlation matrix between the antioxidant variables (DPPH, ABTS, deoxyribose, and iron chelating) and the antibacterial variables indicate perfect correlation, with values of either −1 or 1 (negative or positive), showing linear relationships (Figure 5a). The correlation matrix and heatmap reveal a significant finding: the antioxidant activities (DPPH, ABTS, and desoxyribose) are strongly interrelated. The hierarchical clustering heatmap also shows clear direct associations with the antibacterial activities against EC ATCC 25226, INCQS 00152, PV ATCC 13315, and SE NCTC 12023. Only iron chelating displays clear direct associations with KP ATCC 4352 (Figure 5b). These findings suggest a potential breakthrough: increased antioxidant activity might correspond to lower MIC values (or vice versa) in some cases, a discovery that could have profound implications for our understanding of these activities and their practical applications.

## 4. Discussion

### 4.1. Chemical Profile

Flavonoids are recognized for their potential activity in regulating oxidative stress in cells, while alkaloids are often found in roots, stem bark, wood, and leaves, with a range of biological activities, including antibacterial and antiviral activities, which have been confirmed in in vitro and in vivo tests [18]. A study from Sousa Junior et al. [3] revealed the chemical prospecting of the hydroalcoholic extract of the stem bark of *G. americana*, in which only flababenic tannins and flavonones were present. Secondary metabolites varied throughout the year, exhibiting factors such as seasonality, circadian cycle, plant age, UV radiation, and water stress as some of the causes that directly influence the production of these compounds [19].

According to Rockenbach et al. [20], solvent systems used to extract secondary metabolites have a significant influence on the content of total phenolic compounds and anthocyanins. Their study proved that the phenolic content was best extracted with the solvent acetone (50 and 70%), while the anthocyanins were best extracted with ethanol (50 and 70%). The presence of total phenol content in plant extracts is associated with antioxidant activities, which is explained by their chemical structure and their ability to donate or receive electrons, providing free radical capture. These metabolites are considered strong antioxidants and are involved in antiviral, antimicrobial, antiallergic, and immunomodulatory biological activities [21].

Most of these studies related to *G. americana* have been conducted on fruit with iridoids, which are recurrent for the *Genipa genus* and are involved in the plant’s defense process against insects; moreover, pharmacological potential is demonstrated in various trials for anti-inflammatory, antimicrobial, antiviral, anticancer, and hypoglycemic bioactive properties. Additionally, they are precursors of the monoterpenoid indolic alkaloids involved in cancer treatment, vincristine, and vinblastine [22].

These metabolites showed antioxidant activity, confirming that the Rubiaceae family and the species have properties responsible for sequestering free radicals. Antioxidant substances act by reducing oxidative stress and preventing the harmful effects caused by free radicals, and there are natural antioxidants from vegetables that can be used to prevent various diseases of clinical importance [23]. The presence of caffeic acid, a phenolic derivative, is consistent with the correlation of the genus in the subfamily Ixoridae, and it is in accordance with chemosystematics and botanical positioning. Caffeic acid and other hydroxycinnamic derivatives have biological defense functions against herbivores and pathogens. The ester was isolated from the polar extract of the leaves of *A. macrophylla* (Rubiaceae) [24].

Research using a decoction of the leaves of *Alibertia edulis* (Rubiaceae) showed caffeic acid as well as quercetin, 3-rhamnosyl-(1→6)-galactoside, and ixoside in the chemical composition, revealing 87.09 ± 6.10 mg rutin equivalents (REs) per g of extract for flavonoids and 348.87 ± 2.88 mg gallic acid gram equivalents (GAEs) per g of extract. In a similar study, the caffeic acid content determined was 51.8 mg g^−1^ of the sample. The presence of caffeic acid in the extract under study has been associated with an antiplatelet effect [25].

Phenylethyl caffeate or phenethyl caffeate, caffeic acid phenethyl ester (CAPE) (Figure 1), presents a natural activity that is promising for synthetic antimicrobials and is found as a primary component of temperate propolis. The ester can be obtained in the laboratory by reacting caffeic acid with phenethyl alcohols, which possess hydroxyl groups within the catechol ring, which is associated with many of its biological activities [26].

A study to determine the concentration of total polyphenols in the plant extract revealed 1.535.04 ± 36.05 mg of gallic acid equivalent mL^−1^ and, for phenolic acids, 80.04 ± 4.11 mg of caffeic acid equivalent mL^−1^. These compounds have been reported in the literature to have promising antibacterial activity. According to the same study, minimum inhibitory concentrations (MIC) and minimum bactericidal concentrations (MBC) were satisfactory against *S. aureus* and *E. coli*, showing bacteriolytic action, while scanning electron microscopy revealed morphological changes in the bacterial cells, despite their mild nature [27].

In addition to the biological activities of CAPE, a study by Alfarrayeh et al. [26] showed its high capacity to inhibit planktonic growth and biofilm formation, as well as its ability to partially eradicate mature biofilms of *Candida strains*. The ability of CAPE to inhibit the growth of fluconazole-sensitive and -resistant strains of *C. albicans* and *C. auris* was also noted, especially its ability to rapidly enter *Candida* spp. cells. There are no reports in the literature on the LD_50_ of CAPE in animal models or in normal human cells; however, cytotoxicity was investigated in human multiple myeloma cell lines and the LD_50_ was evaluated at 24, 48, and 72 h, with results of 49.1, 30.6, and 22.5 µg/mL, respectively.

### 4.2. Antibacterial Activity

Pathogenic bacteria can regulate their intermediary metabolism by activating or blocking enzyme synthesis reactions or by altering the structure of their membranes; alternatively, plant antibiotics can be used to treat these agents because they have a different chemical structure to conventional antibiotics [28]. According to Neri et al. [29], Gram-positive bacteria were associated with a greater susceptibility to plant metabolites in comparison to Gram-negative bacteria, due to the structure of their membranes; however, similar studies have shown bacterial growth to be inhibited in different ways, based on the variation in their secondary metabolites and their concentrations, which may have different antibacterial effects. The results of this study revealed satisfactory EECAG MICs for most of the Gram-negative bacterial strains tested, but for caffeic acid, the activity presented was only for *K. pneumoniae*.

The determination of the MIC for the species *Timonius celebicus*, *Psychotria celebica*, and *Gardenia mutabilis* (Rubiaceae), based on the leaves, fruits, and stem, presented weak antibacterial activity for *E. coli*, similar to the result for caffeic acid but different from the results of the extract of this study, which presented an MIC of 256 µg/mL for the same bacteria, demonstrating potential antibacterial activity [30]. The *Morinda citrifolia* L., popularly known as “noni” (Rubiaceae), exhibited activity in vitro against bacterial strains such as *Proteus morgaii*, *Bacillus subtilis*, and *Escherichia coli*. This antibacterial activity conferred on the species, which has been recognized since the 1950, was related to the presence of phenolic compounds present in the fruit such as acubin, *L*-aspersulodieo, and alizarin, while anthraquinonic compounds were conferred in the roots. The phytoconstituent of noni, scopoletin, has been associated with antibacterial activity for *E. coli* [31].

Aminoglycosides are not first-choice antibiotics due to their side effects; however, semi-synthetic derivatives are being investigated in order to improve their pharmacological properties and demonstrate the link between structure and activity, as there are only a few natural and semi-synthetic aminoglycosides in clinical use. These antibiotics act by disrupting bacterial protein synthesis through binding to the A site of the ribosome, a decoding region commonly used in persistent human infections caused by Gram-negative bacteria. Most of them are effective against a wide range of bacteria, although side effects, otoxicity, and nephrotoxicity are some of disadvantages [32].

The use of extracts and other plant products to discover antimicrobial properties has led to a large number of studies intended to find compounds with similar activity to traditional ones, which have reduced toxicity, developed greater efficacy against resistant microorganisms, as well as minimized damage to the environment [33].

The activity of aminoglycosides and their modification by natural products demonstrated their ability to act by reducing their toxicity and increasing their bactericidal activity through a variety of mechanisms, as well as increasing membrane permeability, disrupting enzyme synthesis, and blocking chemical reactions. Not only does it prove efficient, as it reduces the chance of the bacterial strain adapting and developing defense mechanisms, but it can also have synergistic or antagonistic actions [34].

Analyzing Table 2 shows that both the Rubiaceae family and the species under study have secondary metabolites with antibacterial and antioxidant power. The jenipapo stem bark extract showed favorable modulating activity towards aminoglycosides and lincosamides against bacteria of clinical interest such as *P. aeruginosa*. The hydroalcoholic extract of the fruit and the chloroform extract of the leaves of *G. americana* also exhibited antimicrobial activity, in accordance with the results of this study, showing that the species studied from the Rubiaceae family has potential antimicrobial activity [3].

The modifying effect shown by EECGA reveals a relationship with secondary metabolites such as flavonoids, which perform functions such as altering the permeability of the membrane or breaking it, and enhancing the function of antibiotics [40]. The increase in the MIC observed when checking the modifying effect of the extract with amikacin against *S. typhimurium* NCTC 12023 (Figure 2) is due to the antibiotic resistance presented by this species, such as plasmid transport, which enhances the propagation of antibiotic resistance genes through these mediators [36].

Demgne et al. [41] investigated methanol (MeOH), ethyl acetate (EtOAc) extracts, and compounds from the aerial parts of *Psychotria sycophylla* against antibiotic-resistant bacteria. A (MeOH) extract exhibited a broad antibacterial spectrum, acting against 99% of the bacteria used in the study, such as Enterobacter aerogenes EA27 (MIC of 4 µg/mL). This extract provided a notable increase in the efficacy of the antibiotics tested for multidrug-resistant Gram-negative bacteria; furthermore, investigations into the mechanism of action demonstrated that the extract interfered with the bacterial growth process by increasing the lag phase, as well as inhibiting proton pumps.

The synergism observed in this study between jenipapo extract, amikacin, and gentamicin antibiotics may act by reducing the side effects of these aminoglycosides when they are used for prolonged treatment. Furthermore, the extract may act by favoring the influx of the drug into the cell, reducing the difficulties of membrane permeability present in Gram-negative bacteria [34]. A study conducted with the hydroalcoholic extract of *G. americana* stem bark did not show a satisfactory MIC for the bacteria tested; however, it did show potential modifying activity for aminoglycoside antibiotics (amikacin and gentamicin) and lincosamides (clindamycin), preferably for clinically important bacteria such as *P. aeruginosa* [3].

The fact that both the MIC and the modifying potential of the antibiotics showed an activity spectrum for the Gram-negative strains tested is due to the presence of lipophilic compounds acting on the membrane, since Gram-negative bacteria have an outer layer of phospholipids that are structurally lipolysaccharides. This activity is related to compounds such as triterpenes, steroids, anthraquinones, alkaloids, and flavonoids [42]. Antibacterial activity demonstrated by the disk diffusion methodology of the ethanolic extract of the stem of *Rothmannia witti*, a species from the same family, showed that the extract could inhibit the growth of *B. cereus*, *S. aureus*, *Salmonella enteritidis*, *E. coli*, *P. aeruginosa*, and *Salmonella typhimurium*. Halo diameters varied from one species to another [43].

A study with the alcoholic extract of jenipapo bark did not show a favorable MIC for *E. coli*, which was ≥ 1024; however, in the modulation test, there was an agonist effect with the aminoglycosides amikacin and gentamicin. In the same study, no modulatory effect was shown with clindamycin, as this antibiotic has no effect on Gram-negative bacteria [3]. A study by Bona et al. [33] compared the methods for assessing antimicrobial activity, namely disk and well diffusion, and broth microdilution. Agar tests are more commonly used because they provide a simple assessment of the results, such as visualizing the halo; however, disadvantages are presented by the fact that they are labor-intensive because of the preparation of plates and bacterial inoculum. For the microdilution test, it is possible to reduce the space, culture medium, and reagents, managing to perform repetitions and standardized dilutions, thus making the test more reliable.

The resistance of bacteria such as Gram-negative *E. coli* is related to the composition of the bacterial cell wall, which is intrinsically less susceptible to antibacterials. A study conducted with 137 extracts found that none of them showed any activity against this bacterium, diverging from the results of this study [28]. Some interferences may be involved in antimicrobial test results. The use of some extracts does not even show inhibitory activity for the diffusion test; however, the absence of an inhibition zone does not mean that the extract is inactive for the microorganisms tested, instead the result may be related to non-complete diffusion, especially for less polar compounds which diffuse into the culture medium slower [33].

Other species from the Rubiaceae family have been studied, such as *Richardia brasiliensis*, using the ethanolic extract of its aerial parts and roots, which found favorable results against sporulated and non-sporulated Gram-positives and Gram-negatives; similarly, a study with the hydroalcoholic extract of *Hamelia patens* Jacq. revealed potential activity against *E. coli*, *P. aeruginosa*, and *S. aureus* [1]. Research into natural products and synergism with drugs is increasingly in demand, as these studies generally produce satisfactory results for treating infections, as observed in the results of the amikacin and gentamicin antibiotic association with the ethanolic extract of *G. americana* stem bark [44].

### 4.3. Antioxidant Activity

Antioxidants have the function of eliminating free radicals present in the body, as well as acting as a preventative measure against various diseases. This content of antioxidant substances can be influenced by environmental stresses, whether biotic or abiotic. Information on the biochemical composition of plant varieties is important and can potentially be produced in plants [38]. Cell oxidative damage is a risk factor leading to chronic diseases, especially cancer and heart disease, and research into antioxidant compounds is important as they help prevent these diseases. DPPH^●^ is a frequently used chemical method for determining antioxidant capacity, as it is practical, fast, and stable. Studies which investigate the antioxidant activity of jenipapo are frequently conducted only with regard to the fruit [7].

An antioxidant activity test by DPPH^●^ elimination, conducted with the extract of *Paederia foetida* Linn. (Rubiaceae), obtained an approximate IC_50_ value of 75.52 µg/mL, while the IC_50_ value of standard ascorbic acid was 15.78 µg/mL, revealing that it has free radical scavenging activity [35]. Methanolic/acetone, aqueous, and ethanolic extracts of jenipapo fruit investigated by the DPPH^●^ method presented 606.7 µg.mL^−1^ as the best value obtained by the methanolic/acetone combination, demonstrating a low antioxidant capacity and the value of 1092.5 µg.mL^−1^ for the ethanolic extract. However, when compared to other fruits from the Brazilian cerrado, such as pequi, araticum, cagaita, and lobeir, the results are inferior in terms of antioxidant efficacy, with IC_50_ in alcoholic extracts ranging from 148.82 ± 0.98 mg.mL^−1^ to 387.47 ± 8.70 mg.mL^−1^. This study found lower results for the same test using jenipapo bark [7].

A study by Neves et al. [6] compared the different parts of the green fruit of the jenipapo to determine which showed the greatest recovery of the iridoids genipin and geniposide and the antioxidant activity of both parts using the FRAP and DPPH^●^ methods. The results showed the highest genipin content in the endocarp, as well as in the whole fruit, while the highest geniposide content was found in the peel and mesocarp. For the antioxidant tests, the highest FRAP values were found in the endocarp and DPPH^●^ in the mesocarp, as well as the number of total phenols.

Activity should not be based on just one methodology in order to consider a compound as an antioxidant. Methods such as FRAP are an alternative for measuring antioxidant capacity based on the appearance of a bluish color in the solution caused by the reduction of Fe^+3^ to Fe^+2^, which occurs due to the presence of antioxidant compounds; therefore, the greater the amount of Fe^+2^ ion in the solution, the greater its antioxidant potential. The quantitative evaluation of the antioxidant potential of the *Croton argyrophyllus* Kunth extract by the FRAP method showed 167.87 ± 2.88 µM Trolox/g of the sample [29]. In this study, the activity can be seen in Figure 3b,c.

The ABTS^•+^ antioxidant method is frequently used to determine antioxidant activity in food extracts which are suitable for hydrophilic compounds. Hydroethanolic and ethyl acetate extracts of the pulp of the Puruí fruit (Rubiaceae) showed significant antioxidant activity due to decolorization of the ABTS^•+^ cation radical, with results for Trolox-equivalent antioxidant capacity (TEAC) of 28.36 ± 3.7 µM TE/g and 142.26 ± 2.2 μM TE/g [45].

A study with methanolic extracts of the bark and wood of five species of the Rubiaceae family (*Catunaregam tomentosa*, *Haldina cordifolia*, *Mitragyna diversifolia*, *Mitragyna* rotundifolia, and *Morinda coreia*) to investigate the total phenolic (TPC) and total flavonoid (TFC) contents and antioxidant activity showed that the highest TPC and TFC were seen in the wood of *Mitragyna diversifolia* (437.57 ± 9.90 mg GAE g^−1^) and the wood of *Haldina cordifolia* (30.11 ± 0.20 mg QE g^−1^); for DPPH radical scavenging activity, the best results were seen in the bark of *Morinda coreia* (IC_50_ = 360.58 ± 19.28 µg ml^−1^); and for the activity of the reduction of ferric antioxidant power (FRAP), the best results was seen in the wood of M. chorea (IC_50_ = 236.65 ± 1.66 µg ml^−1^) [37].

Investigations of the hydroxyl radical scavenging activity in the species (Rubiaceae) obtained the following results: *Timonius ternifolius* (81.28% ± 4.69), statistically comparable to the ascorbic acid standard (*p* > 0.05); *Psychotria luzoniensis* (62.45% ± 1.49); *Pyrostria subsessilifolia* (39.26% ± 1.09); *Pyrostria triflora* (28.15% ± 1.88); and *Kanapia monstrosa* (the lowest hydroxyl radical scavenging activity at 12.59% ± 3.42). The antioxidant activity capable of scavenging hydroxyl radicals acts by preventing lipid peroxidation and aggressive effects of hydroxyl radicals on cells [39].

The antioxidant activity investigated by the acetone fractions of *Nauclea latifolia* (Rubiaceae) showed the following results for the n-hexane, dichloromethane, ethyl acetate, and n-butanol fractions of leaves, bark, and root bark tested using the DPPH^•^, ABTS^•+^, and FRAP: the n-hexane fraction of leaves showed the highest DPPH^●^ free radical scavenging, with a value of 1011.98 ± 17.01 µM EAA/g; the n-butanol fractions of the peels presented the best ferric reducing power (3056.37 ± 96.66 µmol EAA/g) and the greatest ability to scavenge ABTS^•+^ cation radicals (7031.52 ± 254.98 µmol EAA/g) [21].

The stage of ripeness of the *G. americana* fruit may influence the type of iridoid content. Unripe and ripe fruits were investigated, and the former showed a greater amount of genipin, the iridoid with the greatest contribution to antioxidant activity, investigated using ultrasound-assisted extraction and analyzed by UPLC-DAD-ESI-(-)-QTOF-MS/MS [29]. According to the study by Ngamlai et al. [45], the ABTS^•+^ radical scavenging activity increased proportionally with the concentration of the plant extract; the IC_50_ was 53.165 ± 0.3 μg/mL, while the IC_50_ of standard butylated hydroxytoluene (BHT) was calculated at 8.37 ± 0.21 μg/mL. Concentrations ranging from 10 μg/mL and 100 μg/mL comprised the highest and lowest radical scavenging assignments. Figure 6 shows EECGA’s results in capturing the ABTS^•+^ radical cation. Hydroxyl radicals (OH^●^) originate from Fenton or Haber–Weiss reactions and are classified as one of the most reactive radical classes, causing damage to proteins, lipid membranes, and the DNA molecule. When a substance inhibits the degradation of deoxyribose, it can be related to the inactivation of OH^●^; in the absence of an antioxidant, this substance disrupts the cyclic ring of deoxyribose, producing molondialdehyde (MDA), which, in combination with thiobarbituric acid, results in a pink chromophore that can be absorbed at a wavelength of 530 nm [46].

The formation of OH^●^ can be prevented by substances with antioxidant properties, acting in competition with 2-deoxyribose and/or acting as a Fe^2+^ chelator, providing significant help, especially considering that hydroxyl radicals originate in vivo and are the basis for the development of pathological changes. The presence of antioxidant activity is attributed to the presence of phenolic compounds and flavonoids, secondary metabolites with greater potential for this activity [47]. The main contributions and findings of this manuscript are summarized in Figure 7.

## 5. Conclusions

Chemical prospecting of the *G. americana* bark extract revealed flavonoids and alkaloids, while quantification by HPLC/DAD showed the presence of caffeic acid, chemical compounds responsible for the results. The antibacterial activity was significant for clinically relevant enterobacteria, which can cause conditions ranging from diarrhea to ICU admissions, as well as modifying activity with the aminoglycosides amikacin and gentamicin, demonstrating that the species in question has promising antibacterial and modifying activity.

Even so, *G. americana* can act by reducing the oxidative stress of cells, based on the results obtained by the DPPH^•^ method, capture of the ABTS^•+^ radical cation, and oxidative degradation of deoxyribose, presenting a reduction in Fe^3+^; however, it did not present Fe^2+^ chelating activity. The data obtained can be protected to make *G. americana* a suitable species as an agent with antibacterial and antioxidant functionality in foods, affecting their conservation. To this end, more must be done to prove the possibility of alternatives to improvised preservatives, as there are still few studies in the literature that prove the biological activities of the species and its application, making research a limitation.

## Figures and Tables

**Figure 1 foods-14-00944-f001:**
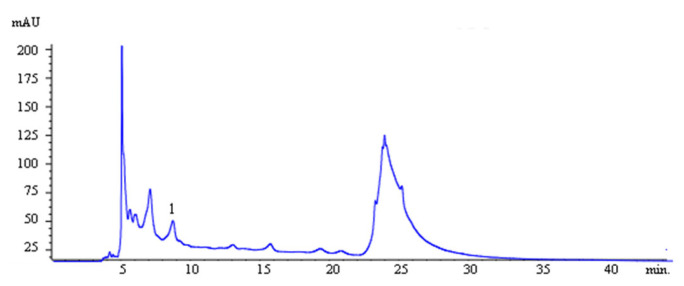
Ethanolic extract of *G. americana* stem bark (EECGA) HPLC profile. Caffeic acid (peak 1).

**Figure 2 foods-14-00944-f002:**
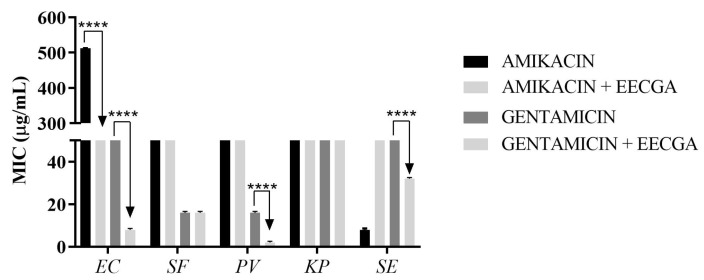
Results of the potential modifier of the antibiotic effect by EECGA on the activity of amikacin and gentamicin against (EC) *E. coli*, (SF) *S. flexneri*, (PV) *P. vulgaris*, (KP) *K. pneumoniae*, and (SE) *S. typhimurium*. Two-way ANOVA followed by Bonferroni post test, using GraphPad Prism 6.0 software. **** *p* < 0.0001. Source: provided by the author.

**Figure 3 foods-14-00944-f003:**
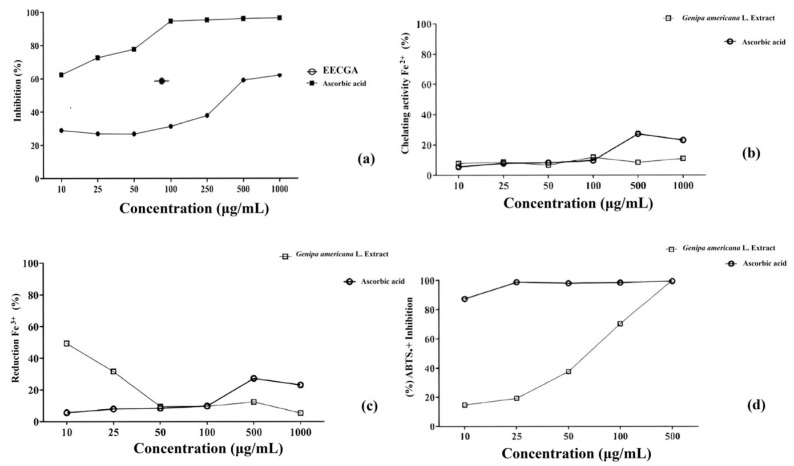
(**a**) Free radical scavenging activity of the EECGA by DPPH^●^ radical inhibition methodology. (**b**) Percentage of Fe^+2^ chelation in the EECGA. (**c**) Reduction of Fe^+3^ in the EECGA. (**d**) Percentage capture of the ABTS^●+^ radical cation in the EECGA. Values expressed as mean ± standard deviation with *p* ˂ 0.05 statistically significant. Source: provided by the author.

**Figure 4 foods-14-00944-f004:**
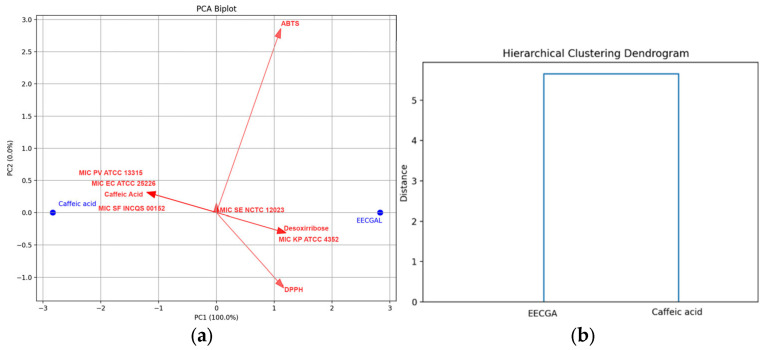
Principal component analysis (PCA) and hierarchical clustering (HCA). (**a**) The biplot illustrates the score plot and the loading plot (indicated by the red arrow) corresponding to the first two principal components. The model accounts for 99.9% of the total variance, primarily represented by PC1. (**b**) Hierarchical clustering analysis, utilizing Euclidean coefficients as the distance measure.

**Figure 5 foods-14-00944-f005:**
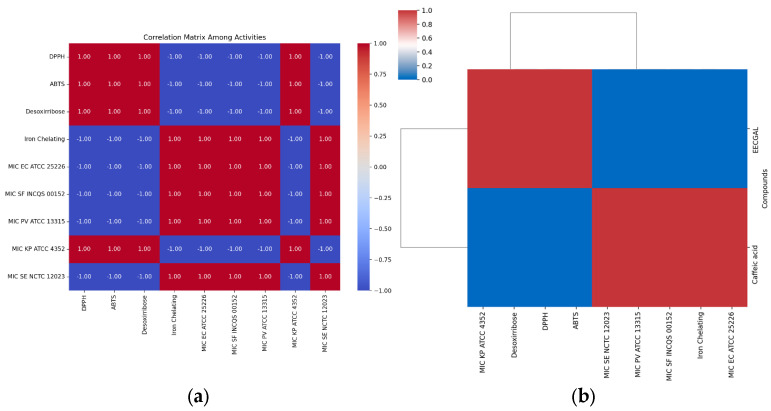
(**a**) A correlation matrix highlighting the relationships between various chemical antioxidants assay and their antibacterial activity, represented in different colors. (**b**) A hierarchical clustering heatmap derived from the entire dataset, focusing on both antioxidant and antibacterial properties.

**Figure 6 foods-14-00944-f006:**
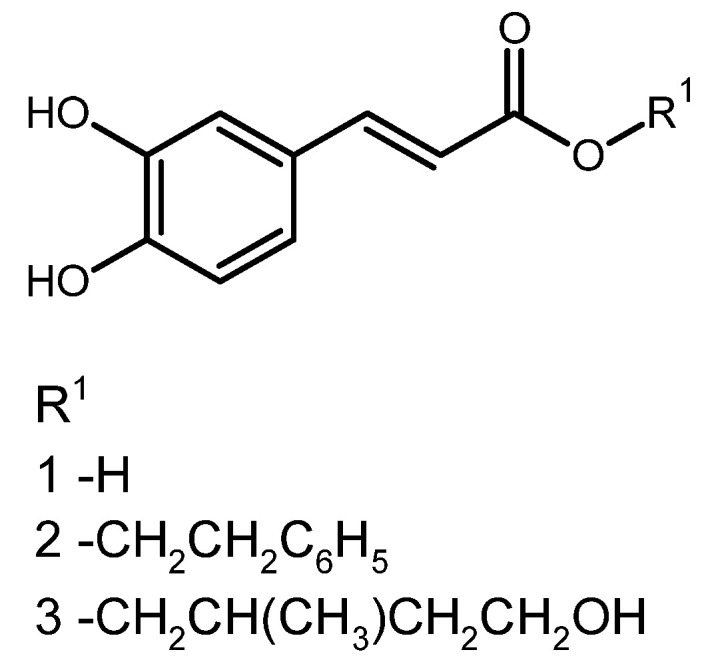
Structural representation of caffeic acid and its phenolic derivatives identified in EECGA.

**Figure 7 foods-14-00944-f007:**
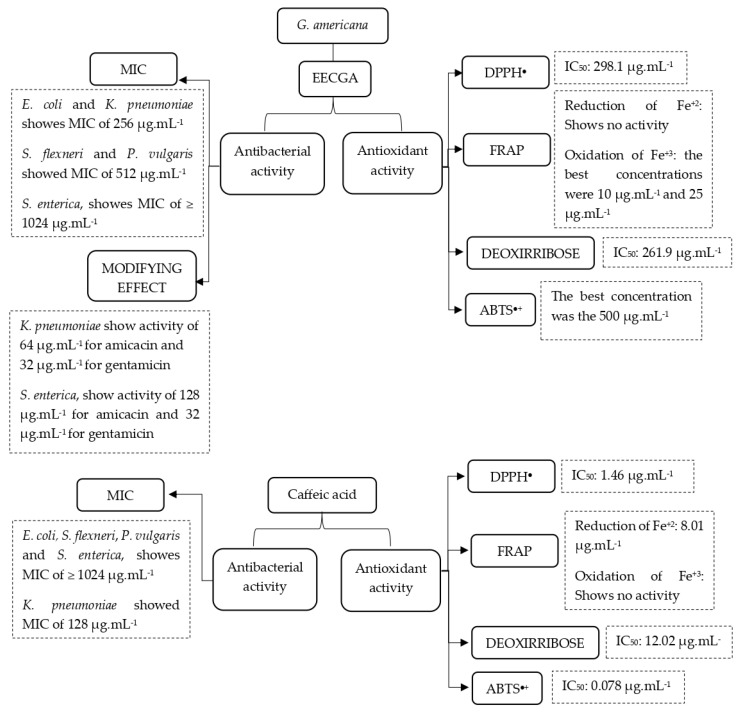
Diagram with the main results of this study.

**Table 1 foods-14-00944-t001:** Classification of secondary metabolites of EECGA.

Classification of Secondary Metabolites												
	**1**	**2**	**3**	**4**	**5**	**6**	**7**	**8**	**9**	**10**	**11**	**12**
EECGA	−	+	+	+	+	+	+	+	+	+	−	+

1: Flababenic tannins; 2: flavones; 3: flavonoids; 4: xanthones; 5: flavonols; 6: chalcones; 7: auroras; 8: catechins; 9: flavonones; 10: catechins; 11: leocoanthocyanins; 12: alkaloids; (+) positive; (−) absent.

**Table 2 foods-14-00944-t002:** Results found by the authors themselves in comparison with the results present in the literature.

	Authors’ Own	Species of the Rubiaceae Family
Chemical Identification	Flavonoids; Alkaloids; Caffeic acid;	Flavonic aglycones, total phenolics, tannins, saponins, coumarin, anthraquinones, triterpenes, and sterols [1]; Flababene tannins and flavones [3]; Iridoids: genipin and geniposide [6]; Total phenols and anthocyanins [7]; Iridoids and flavonoids [22]; Caffeic acid [24]; Caffeic acid [26]; Phenols [35]; Phenols and flavonoids [21];
Extract	Ethanol	Hydroethanol from whole plant maceration [1]; Hydroalcoholic from the peels [3]; Fruit pressurized ethanol [6]; Methanolic/acetonic and ethanolic [7]; Hedroethanol from the leaves [22]; Polar leaf extract [24]; Methanol and ethyl acetate [36]; Hidroetanólic and ethyl acetate [35]; Bark methanolic [37]; n-hexane fraction of leaves; n-butanol fraction of the peels [21];
Species	*G. americana;*	*Richardia brasiliensis* and *Hamenia patens* [1]; *G. americana* [3]; *G. americana* [6] *G. americana* [7]; *A. macrophylla* [22]; *A. edulis* [24]; *Croton argyrophyllus* [28]; *Timonius celebius*, *Psychotria celebia*, and *Gardenia mutabilis* [29]; *Morinda cintrifolia* [30]; *Psychotria sycophylla* [36]; *Paederia foetiza* [38]; *Alibertia edulis* [35]; *Morinda coreia* [39]
Antimicrobial Activity	MIC: *E. coli* and *K. pneumoniae*, 256 µg/mL; *S.flexneri* and *P.vulgaris*, 512 µg/mL	Inhibition halos *S.aureus, Pseudomonas aeruginosa*, and *E. coli* [1]; MIC: *E. coli*, *S. aureus*, and *P. aeruginosa*: ≥1024 µg/mL [3]; MIC 80 values ranged from 12.5 to 100 µg/mL; For strains of *Candida* [28]; MIC: *E. coli*, ≥ 1024 µg/mL [29]; MIC: *Enterobacter aerogenes*, 4 µg/mL [36];
Antioxidant Activity	DPPH^●^: IC_50_ = 298.1 µg/mL^−1^; ABTS^●+^: IC_50_ = 77.25 µg/mL^−1^; FRAP: They did not reach 50% for Fe^3+^ reduction and also did not reach chelation of Fe^2+^; Desoxyribose: IC_50_ = 261.9 µg/mL^−1^	Activity for DPPH^●^ and FRAP, but IC_50_ not provided by the author [6]; DPPH**^●^**: IC_50_ 606.7 µg/mL; 1092.5 µg/mL [7]; FRAP: 167.87 ± 2.88 µM Trolox/g; DPPH**^●^**: IC_50_ 75.52 µg/mL [38]; ABTS**^●+^**: IC_50_ 28.36 ± 3.7 µM TE/g and 142.26 ± 2.2 µM TE/g [35]; DPPH**^●^**: IC_50_ = 360.58 ± 19.28 µg mL^−1^ and FRAP IC_50_ = 236.65 ± 1.66 µg mL^−1^ [37]; DPPH**^●^**: IC_50_ 1011.98 ± 17.01 µM EAA/g, FRAP: 3056.37 ± 96.66 µM EAA/g, and ABTS: 7031.52 ± 254.98 µM EAA/g [39]; DPPH**^●^**: IC_50_ 1011.98 ± 17.01 µmol EAA/g; FRAP: IC_50_ 3056.37 ± 96.66 µM EAA/g, ABTS**^●+^**: IC_50_ 7031.52 ± 254.98 µM EAA/g [21];

## Data Availability

The research materials required for the reproduction of this work are included in the Materials and Methods section. The authors also make this material available upon request to interested researchers.

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
