# Peer review of "Study of the Antioxidant and Antibacterial Effects of Genipa americana L. Against Food Pathogens"

_foods, 2025, doi:10.3390/foods14060944_

Round 1

Reviewer 1 Report

Comments and Suggestions for Authors

Nascimento et al. have aimed to present the chemical profile of ethanolic extract of Genipa americana L. stem bark and evaluate the antioxidant activity and antibacterial activity toward food pathogens. Although the topic is of interest, the authors have failed to present the results appropriately. Thus, the following points need to be addressed elaborately for a possible publication in Foods:

1.        The keyword “food pathogens” should be added.

2.        At the end of the introduction, the formal objectives of this study should be stated.

3.        In all the equations, the “%” on the right side should be removed.

4.        Line 186, the number of decimal points should be controlled as it may be significant to provide so many digits after the decimal point.

5.        Why the authors haven’t provided the complete profile of chemical constituents of G. americana?

6.        Under section 2.1.2, the complete HPLC-DAD analytical conditions should be provided including the column dimensions, flow rate, absorption wavelength etc.

7.        A schematic diagram illustrating the flow of this study along with highlighting the key outcomes should be provided.

8.        It is important to establish correlations between the content of chemical constituents, antioxidant activity and antimicrobial activity by running principal component analysis (PCA) to make lucid elucidations.

9.        A comparison table to compare and discuss the results of parameters obtained in this study with those reported in the literature.

10.     Throughout the manuscript, several 1 or 2-sentence paragraphs should be combined to present a few large paragraphs.

11.     The limitation of this study and future perspective of this work should be highlighted in the conclusion section.

Author Response

Nascimento et al. have aimed to present the chemical profile of ethanolic extract of Genipa americana L. stem bark and evaluate the antioxidant activity and antibacterial activity toward food pathogens. Although the topic is of interest, the authors have failed to present the results appropriately. Thus, the following points need to be addressed elaborately for a possible publication in Foods:

  1. The keyword “food pathogens” should be added.

Response: The keyword “foodborne pathogens” has been added.

  1. At the end of the introduction, the formal objectives of this study should be stated.

Response: Formal objectives have been added.

  1. In all the equations, the “%” on the right side should be removed.

Response: The “%” was removed from the equations.

  1. Line 186, the number of decimal points should be controlled as it may be significant to provide so many digits after the decimal point.

Response: The number of decimal places has been adjusted.

  1. Why the authors haven’t provided the complete profile of chemical constituents of  americana?

Response: All information in Table 1 has been added.

  1. Under section 2.1.2, the complete HPLC-DAD analytical conditions should be provided including the column dimensions, flow rate, absorption wavelength etc.

Ressponse: Methodology has been updated adding the requested data.

  1. A schematic diagram illustrating the flow of this study along with highlighting the key outcomes should be provided.

Response: Diagram added, comprising Figure 7.

  1. It is important to establish correlations between the content of chemical constituents, antioxidant activity and antimicrobial activity by running principal component analysis (PCA) to make lucid elucidations.

Response: Principal component analysis (PCA) was added to the study in Figures 4 and 5.

  1. A comparison table to compare and discuss the results of parameters obtained in this study with those reported in the literature.

Response: Results found by the author himself in comparison with the results present in the literature of Table 2.

  1. Throughout the manuscript, several 1 or 2-sentence paragraphs should be combined to present a few large paragraphs.

Response: The smaller paragraphs were joined together to generate larger paragraphs.

  1. The limitation of this study and future perspective of this work should be highlighted in the conclusion section.

Response: It was organized with more clarity about limitations and future perspectives.

Reviewer 2 Report

Comments and Suggestions for Authors

Dear Authors,

Below is a list of suggestions and comments regarding your manuscript titled "Study of the antioxidant and antibacterial effects of Genipa americana L. against food pathogens":

1.      Line 48: What are the main phenolic components in the stem bark of Genipa americana that were expected to be detected?

2.      Lines 84-88: These two sentences appear to be identical. Please revise for clarity.

3.      HPLC Method Description: Could you specify which solvents were used for solvents A and B? Also, what units were employed to express the results?

4.      Line 105: The ATCC number for Klebsiella pneumoniae is missing. Additionally, the full name of the Salmonella strains should be provided. The correct name is Salmonella enterica subsp. enterica serotype Typhimurium (or simply Salmonella Typhimurium). Please correct this in the manuscript, including lines 25, 198, and 363. Also, what were the positive and negative controls used in the MIC determination?

5.      Antibacterial Activity: Why was the evaluation of antibacterial activity limited to gram-negative bacteria?

6.      Line 130: In formula (1), the term "Abs.Control" is used, while in the explanation below, "Abs.Negative control" is mentioned. Please unify the terminology.

7.      Deoxyribose Oxidative Degradation: Could you specify the units used for determining deoxyribose oxidative degradation?

8.      Chemical Prospecting (Lines 181-183): It is suggested that the findings related to the chemical prospecting of the ethanolic extract of G. americana bark be presented in a table, clearly showing which groups of chemical compounds yielded negative results.

9.      Chromatogram: Please provide a chromatogram of the analyzed extract from the bark of G. americana. Were there any unidentified peaks in the analysis?

10.  Bacterial Strains: Did you perform the antibacterial activity analysis against all the gram-negative bacteria used in the study? If so, please provide the results for all strains. If not, kindly explain why only Klebsiella pneumoniae and Salmonella Typhimurium were selected.

11.  Line 210-211: The sentence referring to Croton argyrophyllus Kunth needs better integration into the text. For example, you could introduce it as: "Croton argyrophyllus Kunth, another plant species from northeastern Brazil, was investigated for its antioxidant potential by the FRAP method, yielding 167.87 ± 2.88 μM Trolox/g of the sample."

12.  IC50 Values of Catechin: Where are the IC50 values for catechin, as determined by the DPPH method? You mentioned using ascorbic acid and catechin as controls—please clarify this in the manuscript.

13.  Acronyms/Abbreviations: Please follow the instructions for authors regarding the use of acronyms/abbreviations. They should be defined the first time they appear in the abstract, main text, and figures or tables. For example, the abbreviation "EECGA" should be defined upon the first mention in the figure, as well. Additionally, the abbreviation "EECGAL" appears in Figure 2—please revise this. Also, in Figures 2 and 3, "EECGA" is used, whereas in Figures 4, 5, and 6, "Genipa americana L. Extract" is used. Please standardize this throughout the manuscript.

14.  Figure Presentation: It is suggested that Figures 3, 4, 5, and 6 be combined into a single figure for better clarity of the results.

15.  Section 4.1 - Chemical Profile: This section requires improvement as it lacks a clear flow of discussion and comparison with existing literature. For example, lines 292-298 should be better integrated into the study context. Additionally, this section focuses on chemical profiling and may be better placed under Section 4.3 (Antioxidant Activity). Line 269 also requires clarification—please specify which test was used.

16.  Italicization: Please italicize "in vitro" and "in vivo" in Line 233. Additionally, correct "E. coli" (Lines 333, 394) and "P. aeruginosa" (Line 355).

17.  Figure 1: Figure 1 appears after Figure 6. Please revise the order to ensure proper sequencing.

18.  Section 4.3 - Antioxidant Activity: This section would benefit from a comparison of your results with relevant literature data.

19.  Discussion Section: Sentences such as "In this study, the activity can be seen in (Figure 4)" and "Figure 7 shows EECGA's results in capturing the ABTS•+ radical cation" (Lines 460-476) should be reworded. These points are already presented in the results section, so the findings should be explained and compared with the existing literature in the discussion. Please revise this accordingly.

Author Response

Comments and Suggestions for Authors

Dear Authors,

Below is a list of suggestions and comments regarding your manuscript titled "Study of the antioxidant and antibacterial effects of Genipa americana L. against food pathogens":

  1. Line 48: What are the main phenolic components in the stem bark of Genipa americanathat were expected to be detected?

Response: Studies with extracts from jenipapo bark are scarce, most studies are with the fruit, aiming to limit the comparison of these studies.

  1. Lines 84-88: These two sentences appear to be identical. Please revise for clarity.

Response: The two sentences have been adjusted and revised.

  1. HPLC Method Description: Could you specify which solvents were used for solvents A and B? Also, what units were employed to express the results?

Response: The methodology was adjusted and the requested information was added.

  1. Line 105: The ATCC number for Klebsiella pneumoniaeis missing. Additionally, the full name of the Salmonella strains should be provided. The correct name is Salmonella enterica subsp. enterica serotype Typhimurium (or simply Salmonella Typhimurium). Please correct this in the manuscript, including lines 25, 198, and 363. Also, what were the positive and negative controls used in the MIC determination?

Response: The ATCC number for Klebsiella pneumoniae was added and the name Salmonella typhimurium was used replacing it as it was previously.

  1. Antibacterial Activity: Why was the evaluation of antibacterial activity limited to gram-negative bacteria?

Response: The species is popularly used for gastrointestinal problems caused by enterobacteria and this.

  1. Line 130: In formula (1), the term "Abs.Control" is used, while in the explanation below, "Abs.Negative control" is mentioned. Please unify the terminology.

Response: Terminology has been standardized.

  1. Deoxyribose Oxidative Degradation: Could you specify the units used for determining deoxyribose oxidative degradation?

Response: Corrections highlighted in the text.

  1. Chemical Prospecting (Lines 181-183): It is suggested that the findings related to the chemical prospecting of the ethanolic extract of G. americanabark be presented in a table, clearly showing which groups of chemical compounds yielded negative results.

Response: The table with all the necessary information has been added.

  1. Chromatogram: Please provide a chromatogram of the analyzed extract from the bark of G. americana. Were there any unidentified peaks in the analysis?

Response: The chromatogram has been added.

  1. Bacterial Strains: Did you perform the antibacterial activity analysis against all the gram-negative bacteria used in the study? If so, please provide the results for all strains. If not, kindly explain why only Klebsiella pneumoniaeand Salmonella Typhimurium were selecte.

Response: Yes, the results have been added to Figure 2.

  1. Line 210-211: The sentence referring to Croton argyrophyllus Kunth needs better integration into the text. For example, you could introduce it as: "Croton argyrophyllus Kunth, another plant species from northeastern Brazil, was investigated for its antioxidant potential by the FRAP method, yielding 167.87 ± 2.88 μM Trolox/g of the sample."

Response: The phrase has been replaced as suggested.

  1. IC50 Values of Catechin: Where are the IC50 values for catechin, as determined by the DPPH method? You mentioned using ascorbic acid and catechin as controls—please clarify this in the manuscript.

Response: Catechin was not used in this method, only the positive control (ascorbic acid), therefore, the part that mentioned catechin was removed from the methodology.

  1. Acronyms/Abbreviations: Please follow the instructions for authors regarding the use of acronyms/abbreviations. They should be defined the first time they appear in the abstract, main text, and figures or tables. For example, the abbreviation "EECGA" should be defined upon the first mention in the figure, as well. Additionally, the abbreviation "EECGAL" appears in Figure 2—please revise this. Also, in Figures 2 and 3, "EECGA" is used, whereas in Figures 4, 5, and 6, "Genipa americanaL. Extract" is used. Please standardize this throughout the manuscript.

Response: According to the suggestions, the acronyms were standardized.

  1. Figure Presentation: It is suggested that Figures 3, 4, 5, and 6 be combined into a single figure for better clarity of the results.

Response: The figures were combined into one, comprising figure 3, with sections 3 (a, b, c and d).

  1. Section 4.1 - Chemical Profile: This section requires improvement as it lacks a clear flow of discussion and comparison with existing literature. For example, lines 292-298 should be better integrated into the study context. Additionally, this section focuses on chemical profiling and may be better placed under Section 4.3 (Antioxidant Activity). Line 269 also requires clarification—please specify which test was used.

Response: The suggestions were reviewed and adapted to what is requested, however, there are limitations in points such as the literature not presenting many studies that can be compared to the species Genipa americana and the Rubiaceae family that fit the idea of ​​the article.

  1. Italicization: Please italicize "in vitro" and "in vivo" in Line 233. Additionally, correct "E. coli" (Lines 333, 394) and "P. aeruginosa" (Line 355).

Response: They have been corrected and placed in italics.

  1. Figure 1: Figure 1 appears after Figure 6. Please revise the order to ensure proper sequencing.

Response: Sequencing has been standardized in the correct order.

  1. Section 4.3 - Antioxidant Activity: This section would benefit from a comparison of your results with relevant literature data.

Response: Paragraphs with antioxidant data were added to better establish a relationship with the results.

  1. Discussion Section: Sentences such as "In this study, the activity can be seen in (Figure 4)" and "Figure 7 shows EECGA's results in capturing the ABTS•+ radical cation" (Lines 460-476) should be reworded. These points are already presented in the results section, so the findings should be explained and compared with the existing literature in the discussion. Please revise this accordingly.

Response: The points were clarified and are highlighted in the text.

Round 2

Reviewer 1 Report

Comments and Suggestions for Authors

The authors have satisfactorily addressed all the comments raised by reviewers and substantially improved the overall quality of the article. Therefore, I recommend accepting this article for publication in Foods.

Reviewer 2 Report

Comments and Suggestions for Authors

Dear Authors,

Thank you for incorporating my comments and suggestions. The manuscript is well-written, and I can see significant improvements.

For future articles, however, I would like to ask that you highlight the changes made in the revised version. This will make it much easier to track the adjustments and ensure clarity during the review process.